

# Proximate grassland and shrub-encroached sites show dramatic restructuring of soil bacterial communities

Xingjia Xiang[1], Sean M. Gibbons[2], He Li[3], Haihua Shen[4] and Haiyan Chu[5]

[1] Anhui Province Key Laboratory of Wetland Ecological Protection and Restoration, School of Resources and Environmental Engineering, Anhui University, Hefei, China
[2] Institute for Systems Biology, Seattle, WA, USA
[3] School of Geography, Geomatics and Planning, JiangSu Normal University, Xuzhou, China
[4] State Key Laboratory of Vegetation and Environmental Change, Institute of Botany, Chinese Academy of Sciences, Beijing, China
[5] State Key Laboratory of Soil and Sustainable Agriculture, Institute of Soil Science, Chinese Academy of Sciences, Nanjing, China

Corresponding author
Haiyan Chu, hychu@issas.ac.cn

## ABSTRACT

**Background:** Changes in aboveground community composition and diversity following shrub encroachment have been studied extensively. Recently, shrub encroachment was associated with differences in belowground bacterial communities relative to non-encroached grassland sites hundreds of meters away. This spatial distance between grassland and shrub sites left open the question of how soil bacterial communities associated with different vegetation types might differ within the same plot location.

**Methods:** We examined soil bacterial communities between shrub-encroached and adjacent (one m apart) grassland soils in Chinese Inner Mongolian, using high-throughput sequencing method (Illumina, San Diego, CA, USA).

**Results:** Shrub-encroached sites were associated with dramatic restructuring of soil bacterial community composition and predicted metabolic function, with significant increase in bacterial alpha-diversity. Moreover, bacterial phylogenic structures showed clustering in both shrub-encroached and grassland soils, suggesting that each vegetation type was associated with a unique and defined bacterial community by niche filtering. Finally, soil organic carbon (SOC) was the primary driver varied with shifts in soil bacterial community composition. The encroachment was associated with elevated SOC, suggesting that shrub-mediated shifts in SOC might be responsible for changes in belowground bacterial community.

**Discussion:** This study demonstrated that shrub-encroached soils were associated with dramatic restructuring of bacterial communities, suggesting that belowground bacterial communities appear to be sensitive indicators of vegetation type.

Our study indicates that the increased shrub-encroached intensity in Inner Mongolia will likely trigger large-scale disruptions in both aboveground plant and belowground bacterial communities across the region.

## INTRODUCTION

Increased cover, abundance and dominance of shrub species in grasslands have been widely reported, with 10–20% of arid and semiarid grassland area undergoing shrub encroachment across the world (*Van Vegten, 1984*; *Jackson et al., 2002*; *Maestre et al., 2009*). Multiple factors appear to trigger shrub encroachment, including grazing pressure (*Coetzee et al., 2008*), climate change (i.e., global warming, elevated $CO_2$, nitrogen deposition; *Archer, Schimel & Holland, 1995*) and wildfire frequency (*Scholes & Archer, 1997*). Around 330 million ha of grassland were subject to shrub invasion in xeric western states of United States (*Knapp et al., 2008*). A total of 13 million ha of savanna are undergoing shrub encroachment in South Africa (*Eldridge et al., 2011*). Moreover, similar conditions were demonstrated in many other areas of the world (e.g., Eurasian and Australian grasslands; *Zhang et al., 2006*; *Rivest et al., 2011*; *Chen et al., 2015*). Shrub encroachment significantly affects the livestock industry, which also has important ecological repercussions in arid and semiarid grasslands.

Shrub encroachment into native grassland results in a loss of biodiversity that can affect ecosystem functioning (*Throop & Archer, 2008*). Areas undergoing encroachment are characterized by patchy vegetation, with clusters of shrubs and areas dominated by grasses. Shrub and grass patches differ in above-ground community composition, overall primary productivity, plant allocation, and rooting depth (*Trumbore, 1997*; *Briggs et al., 2005*; *McClaran et al., 2008*; *Meyer, Wiegand & Ward, 2009*), leading to the long-term profound effects of encroachment on grassland ecosystems, including changes in soil erosion, soil moisture (SM), soil carbon, soil pH, energy cycling, soil aeration, soil nitrogen contents, and soil faunal communities (*Lett & Knapp, 2003*; *Smith & Johnson, 2003*; *Breshears, 2006*; *Knapp et al., 2008*; *McKinley & Blair, 2008*). The impacts induced by encroachment are not always coincident, sometimes leading to a decrease (*Gómez-Rey et al., 2013*) or an increase (*Soliveres & Eldridge, 2014*) in aboveground plant productivity. Shrub encroachment is often related to soil nutrient accumulation ("islands of fertility"; *Reynolds et al., 1999*; *Peng et al., 2013*) due to litterfall and nitrogen fixation (*Schlesinger et al., 1990*; *Hibbard et al., 2001*).

It is plausible that complicated feedback mechanisms present among aboveground vegetation, belowground properties, and microbial communities (*Hart et al., 2005*). Soil microorganisms play crucial roles in belowground ecosystems, serving as catalysts for nutrient transformations, forming mutualistic relationship with plants to improve host health, and working as engineers to maintain soil structure (*Hart et al., 2005*; *Paul & Clark, 1996*). Shrub encroachment triggers large shifts in plant and soil properties, which may directly and indirectly affect soil microbial communities. Soil properties, such as soil carbon content (*Zhang et al., 2014*) and pH (*Griffiths et al., 2011*) significantly affect microbial community structure. Plant litterfall and root exudates provide nutrients to feed soil heterotrophic microbes (*Staddon et al., 2003*). Previous studies have found that revegetation significantly affected soil microbial biomass and community structure (*Yannarell, Menning & Beck, 2014*; *Bragazza et al., 2015*). Shrub encroachment significantly altered soil microbial communities, soil respiration, extracellular enzyme

activity, and denitrification potential in subtropical marshes (*Ho & Chambers, 2019*). Previously, we found dramatic shifts in soil bacterial communities associated with shrub encroachment relative to distant grassland soils (i.e., >500 m), without exploring associations between vegetation type and soil properties within the same sampling location (*Xiang et al., 2018*).

In China, shrubs have occupied more than 5.1 million ha grassland in Inner Mongolian of China (*Chen et al., 2015*). A better understanding of bacterial community structure in shrub-encroached soils is crucial for clarifying the influence of encroachment on grassland ecosystem functioning. In this study, we focus on soil bacterial community composition within shrub-dominated and adjacent grassland-dominated patches (one m apart) in the same sampling site. In particular, we addressed two main questions: (i) how encroachment affects soil bacterial community composition and diversity; and (ii) what are the main factors driving soil bacterial communities following shrub encroachment.

## MATERIALS AND METHODS

### Site description and sample collection

The study area was selected in a high-density shrub-encroached grassland (42°57′N, 112°43′E; 1,208 m; Fig. S1), located in Inner Mongolia, China. The average annual temperature is 5.1 °C and the mean precipitation is 195 mm in this region (*Chen et al., 2015*). The dominated grass is *Cleistogenes songorica* across the region, but *Caragana microphylla* is encroaching (*Chen et al., 2015*). Soil samples were collected on the 10th of August, 2016. We identified ten shrub-encroached sample plots to include in this study. The selected sites were more than 500 m away from each other. At each site (10 × 10 m), the encroachment soils were sampled under five shrub patches (the nearest to the four vertices and the center of a plot) with 0–10 cm depth and mixed as one sample. The control non-encroached soils were collected one m away from the five shrub canopies with 0–10 cm depth and mixed as one sample (Fig. S1). In total, 10 from control grassland soils and 10 from adjacent encroached soils were collected for further study. The soils were fully mixed and sieved, and then transported refrigerated to the lab within 24 h. The soils were divided into two parts: one part was stored at 4 °C for biogeochemical analysis and the other was stored at −20 °C for DNA extraction.

### Sample pretreatment

Measurement of soil properties, DNA extraction, and amplicon library preparation are described in the Supplemental Information.

### Processing of sequence data

The raw data were processed by QIIME (v.1.9.0; *Caporaso et al., 2010*). The sequences were clustered into operational taxonomic units (OTUs; 97% identity) with UCLUST (*Edgar, 2010*). Chimeric and singleton OTUs were removed prior to downstream analysis. The default setting was used to select the representative sequence (i.e., most abundant sequence) for each OTU, which was assigned taxonomic annotations using the UCLUST (*Edgar, 2010*) and aligned by PyNAST (*Caporaso et al., 2010*). To normalize for sampling

depth, random subsets of 26,000 reads per sample (the lowest sequence read depth across the study) were used to calculate bacterial alpha- and beta-diversities.

## Statistical analysis

Phylogenetic diversity (PD) was estimated by Faith's index (*Faith, 1992*). Pairwise *t*-test was performed to show differences in relative abundance of dominant bacterial phyla and alpha-diversity. Pearson correlation was used to test relationships between bacterial alpha-diversity and soil properties. Linear discriminant analysis effect size (LEfSe) was used to identify bacterial taxa that differed significantly between treatments (default setting; *Segata et al., 2011*). Non-metric multidimensional scaling and Analysis of Similarity (ANOSIM; permutations = 999) were performed to distinguish the differences in bacterial community composition between treatments by using the vegan package (v.2.0-2) in R software. The correlation between variables (i.e., soil properties and spatial distance) and soil bacterial community composition were analyzed by Mantel tests (permutations = 999). Multicollinearity of soil properties was tested by the variance inflation factor (VIF; *Zuur, Leno & Elphick, 2010*), and those properties with the VIF values < 3 were selected for canonical correspondence analysis (CCA).

The nearest taxon index (NTI) and beta nearest taxon index (betaNTI) were performed using the picante package (*Purcell et al., 2007*) and Phylocom 4.2 (*Hardy, 2008*), respectively, to analyze soil bacterial phylogenetic structure. The NTI measures the mean nearest taxon distance among individuals to estimate the phylogenetic dispersion of the community (*Webb, 2000*). More positive or negative NTI values indicate phylogenetic clustering or overdispersion, respectively (*Webb, 2000*). BetaNTI values between −2 and 2 suggested stochastic process (neutral assembly) while the values above 2 or below −2 indicated deterministic processes (niche assembly, *Stegen et al., 2012*). Co-occurrence networks were generated in R using the "WGCNA" package (*Langfelder & Horvath, 2012*). We adjusted all *P*-values (cutoff as 0.001) by using the Benjamini and Hochberg false discovery rate for multiple testing (*Benjamini, Krieger & Yekutieli, 2006*). The network nodes defined as network hubs ($z$-score > 2.5; $c$-score > 0.6), module hubs ($z$-score > 2.5; $c$-score < 0.6), connectors ($z$-score < 2.5; $c$-score > 0.6), and peripherals ($z$-score < 2.5; $c$-score < 0.6) referring to their roles in network structure (*Poudel et al., 2016*). Network hubs are those OTUs that are highly connected both in general and within a module. Module hubs and connectors are OTUs that are highly connected only within a module and only link modules, respectively. Peripherals are defined as OTUs that have few links to other species. The bacterial metabolic function was predicted by phylogenetic investigation of communities by reconstruction of unobserved states (PICRUSt) according to KEGG database (*Langille et al., 2013*).

# RESULTS

## Soil chemistry

Compared to non-encroached grassland soils, shrub-encroached soils were associated with higher content of $NO_3^-$, total nitrogen (TN), total carbon (TC), dissolved organic carbon (DOC), soil organic carbon (SOC), and total phosphorus (Table S1). However,

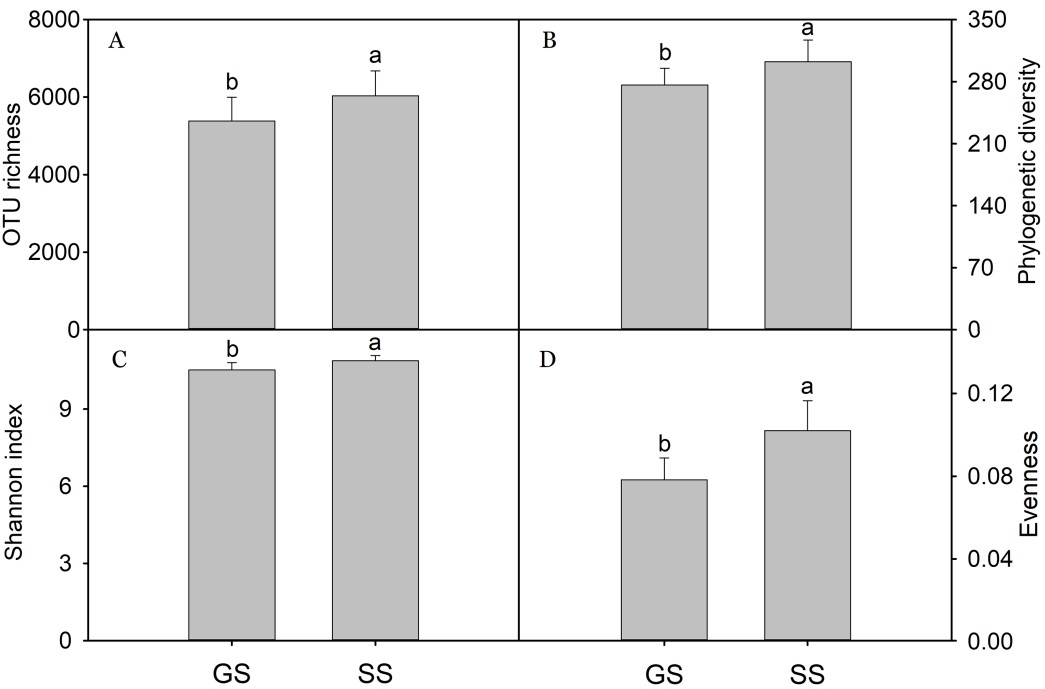

**Figure 1 Soil bacterial diversity.** Bacterial alpha-diversity calculated at a rarefaction depth of 26,000 randomly selected sequences per sample. (A) bacterial OTU richness; (B) bacterial phylogenetic diversity; (C) bacterial Shannon index; (D) bacterial evenness. Bars represent mean; error bars denote standard deviation; letters above bars represents significant differences from pairwise *t*-test (*P* < 0.05). GS: control grassland soil; SS: shrub-encroached soil.

shrub encroachment showed little effect on other soil properties, such as soil pH, $NH_4^+$ content and SM relative to control in this study (Table S1).

## Bacterial alpha-diversity

A total of 966,631 quality bacterial sequences was obtained with 26,037–68,261 (mean 48,332) sequences per sample. In this study, bacterial alpha-diversity included OTU richness, Shannon index, evenness, and PD, which was calculated by randomly selected subsets of 26,000 reads per sample. Generally, encroached sites had significant higher alpha-diversity relative to grassland sites (Fig. 1). Bacterial OTU richness was positively correlated with $NO_3^-$, DOC, TC, TP, and SOC; PD was positively correlated with $NO_3^-$, DOC, TC, and SOC; the Shannon index was positively correlated with $NO_3^-$, TC, TP, and SOC; evenness was positively correlated with $NO_3^-$, TN, TC, and SOC (Table 1).

## Bacterial community structure

The dominant soil bacterial phyla (i.e., relative abundance > 1%) across all samples were Actinobacteria (27.3%), Acidobacteria (23.1%), Proteobacteria (23.0%), Chloroflexi (6.0%), Planctomycetes (4.7%), Gemmatimonadetes (2.8%), Firmicutes (2.7%), Bacteroidetes (2.6%), and Nitrospirae (2.4%) (Fig. S2). Compared to control grassland soils, the relative abundance of Proteobacteria showed significantly lower in encroached sites (Fig. S3). Compared to control, encroachment was associated with

**Table 1 The correlation between bacterial alpha-diversity and soil properties.**

| Variables | OTU | PD | Shannon | Evenness |
|---|---|---|---|---|
| Soil pH | 0.276 | 0.327 | 0.077 | −0.164 |
| Soil moisture (%) | 0.167 | 0.109 | 0.315 | −0.045 |
| $NH_4^+$-N (mg/kg) | −0.300 | −0.146 | 0.276 | 0.413 |
| $NO_3^-$-N (mg/kg) | **0.502***  | **0.532*** | **0.475*** | **0.563*** |
| Dissolved organic C (mg/kg) | **0.471*** | **0.483*** | 0.402 | 0.400 |
| Dissolved organic N (mg/kg) | 0.131 | 0.148 | 0.176 | 0.138 |
| Total carbon (mg/g) | **0.486*** | **0.495*** | **0.491*** | **0.485*** |
| Total nitrogen (mg/g) | 0.389 | 0.394 | 0.362 | **0.474*** |
| Total phosphorus (mg/g) | **0.462*** | 0.421 | **0.493*** | 0.255 |
| Soil inorganic carbon (mg/g) | −0.058 | 0.019 | −0.014 | 0.248 |
| Soil organic carbon (mg/g) | **0.582*** | **0.586*** | **0.574*** | **0.601*** |

Notes:
  Significant correlations are shown in bold ($P < 0.05$).
  * $P < 0.05$; OTU, operational taxonomic unit; PD, phylogenetic diversity.

higher relative abundance of Chloroflexi and Nitrospirae (Fig. S3). LEfSe analysis showed that bacteria in one phylum (i.e., Proteobacteria), five classes (i.e., *Acidobacteriia, ML635J_21, vadinHA49, Solibacteres*, and *Gammaproteobacteria*) and 15 orders (i.e., *Acidobacteriales, Solibacterales, Planctomycetales, Caulobacterales, Rhodospirillales, Burkholderiales,* etc) were significantly more abundant in control soils. Bacteria from two phyla (i.e., Nitrospirae and Armatimonadetes), two classes (i.e., *Nitrospira* and *Chloroflexi*) and 11 orders (i.e., *Gaiellales, Roseiflexales, Nitrospirales, Syntrophobacterales, Desulfovibrionales,* etc) were significantly more abundant in shrub-encroached soils (Fig. 2).

Significant differences in soil bacterial community compositions were found between shrub-encroached and grassland sites (ANOSIM: $P = 0.001$; Fig. 3). The NTI values showed positive (i.e., >0; $P = 0.001$) for all samples, indicating that bacterial phylogenetic structure showed clustering in both encroached and control soils (Fig. 4). Almost all betaNTI scores for bacterial communities were below −2, which suggested that deterministic assembly dominated soil bacterial community dynamics in both grassland and shrub-encroached soils (Fig. 4). A correlation network was built at bacterial genus level. There was a larger proportion of positive than negative correlations between genera in soils (Fig. S4A). Compared to grassland soils, shrub-encroached soil showed higher proportion of correlation network hubs (Fig. S4B), suggesting that bacterial community in shrub-encroached soils might be more interconnected than grassland soils.

Mantel tests demonstrated that soil bacterial community composition showed significant correlation with soil pH, SM, $NO_3^-$, DOC, TC, TP, and SOC (Table 2; $P < 0.05$ in all cases). Among these variables, SOC content ($P = 0.002$) had the strongest correlation with soil bacterial community composition. However, spatial distance showed little correlation with bacterial community composition ($P = 0.181$; Table 2). CCA further demonstrated that SOC was the primary driver affecting soil bacterial community composition (Fig. S5).

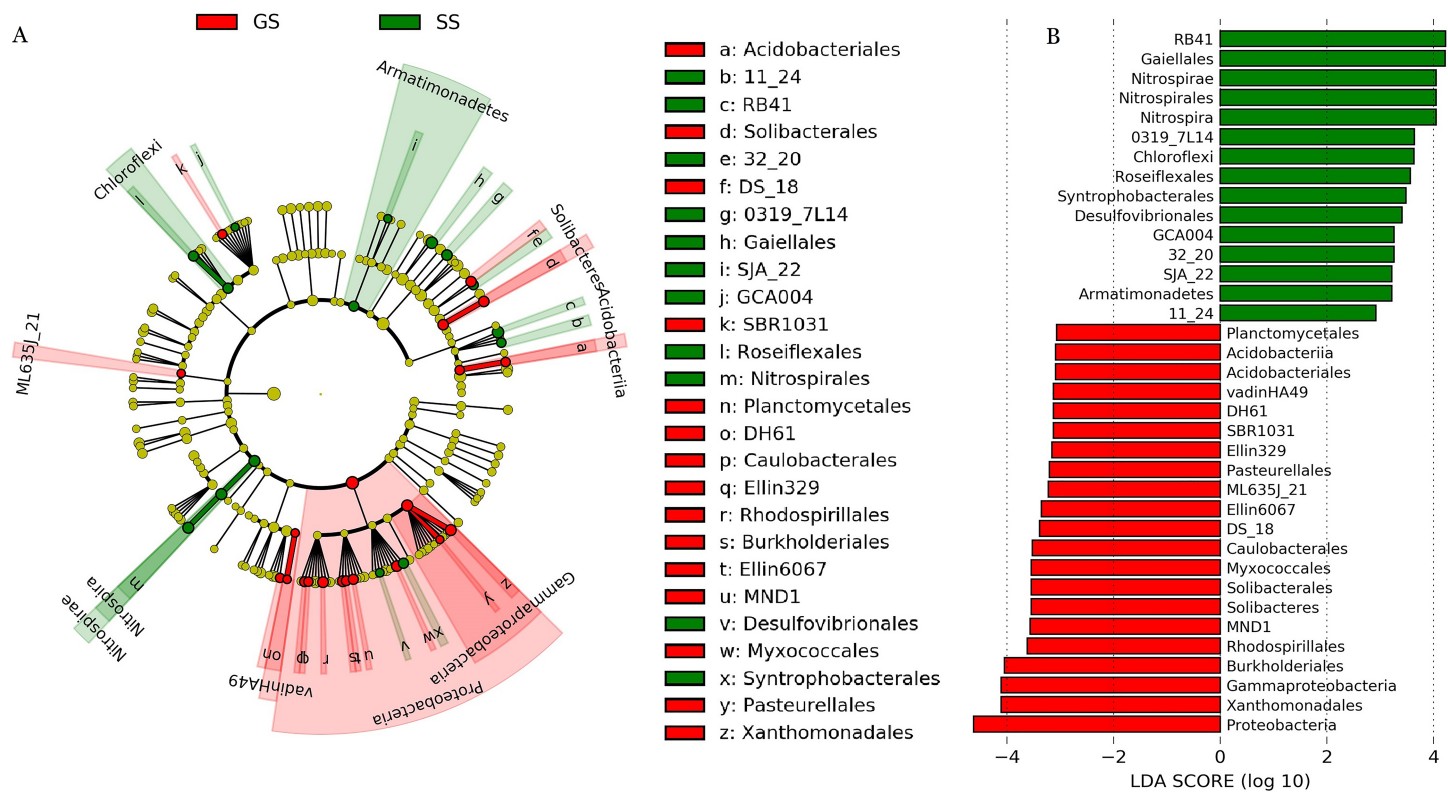

**Figure 2** **LEfSe analysis of soil bacterial biomarkers associated with vegetation type.** (A) cladogram representing the taxonomic hierarchical structure of the phylotype biomarkers identified between two vegetation types. Each filled circle represents one biomarker. Red, phylotypes statistically overrepresented in grassland soil; green, phylotypes overrepresented in shrub-encroached soil; yellow, phylotypes for which relative abundance is not significantly different between the two vegetation types. (B) Identified phylotype biomarkers ranked by effect size and the alpha value was <0.05.

## The predicted metabolic function

The metabolic function of bacterial community was predicted by PICRUSt. A total of 328 predicted functional genes were detected in this study. More than 89% of total sequences belonged to categories of metabolism (52.2%), genetic information processing (15.8%), environmental information processing (13.3%), and organismal systems (8.35%) in soils, according to the KEGG database. Compared to controls, shrub encroachment was associated with significant differences in potential functions of the soil bacterial community (Fig. S6). Metabolism of cofactors and vitamins, energy metabolism, glycan biosynthesis and metabolism, enzyme families, and nucleotide metabolism were enriched in grassland soil, while xenobiotics biodegradation and metabolism, lipid metabolism, metabolism of terpenoids and polyketides, amino acid metabolism, and carbohydrate metabolism were enriched in shrub-encroached soils (Fig. 5). The relative abundances of sequences associated with cell motility, environmental adaptation, signal transduction, and protein folding, sorting and degradation were enriched in grassland soils (Fig. 5). The sequences related to cell growth and death, transport and catabolism, nervous system, membrane transport, and transcription were enriched in shrub-encroached soils (Fig. 5).

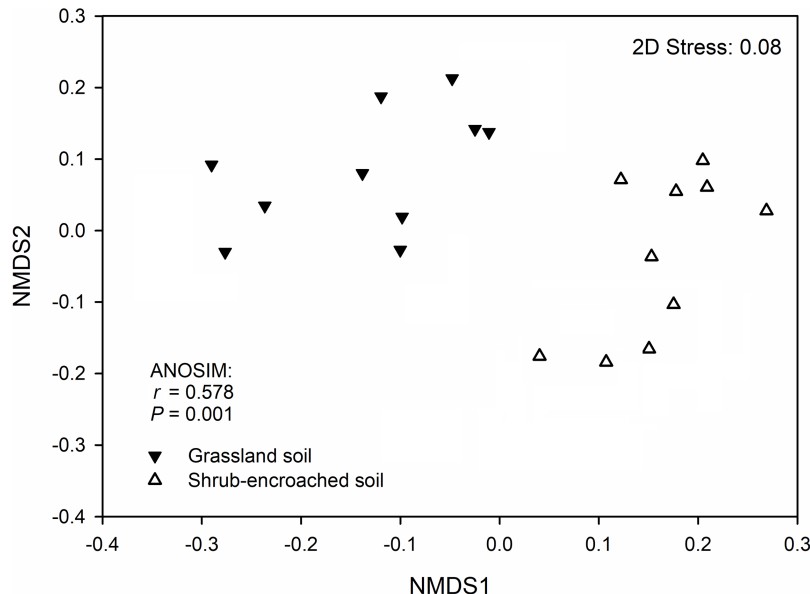

**Figure 3 Non-metric multidimensional scaling (NMDS) plot.** Non-metric multidimensional scaling (NMDS) plot showing bacterial community composition in control grassland and shrub-encroached soils of Inner Mongolia.                             

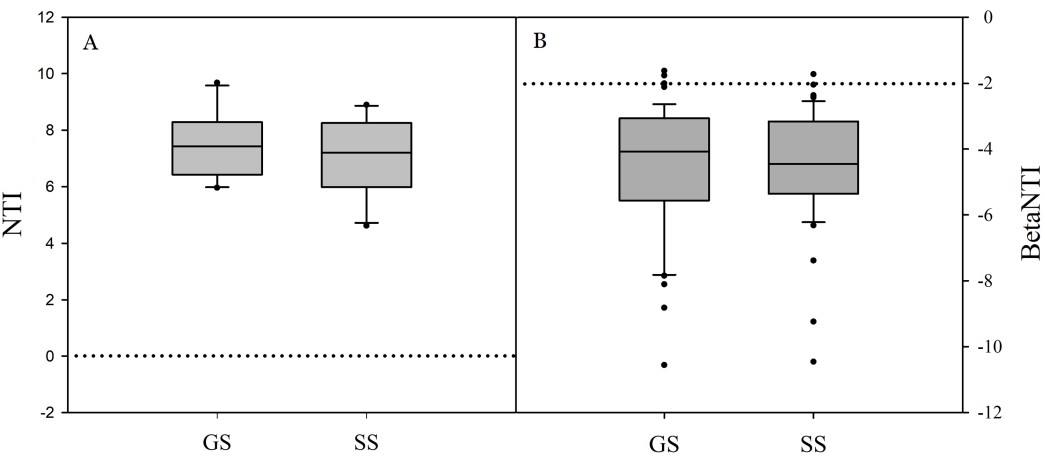

**Figure 4 The values of nearest taxon index (NTI; A) and beta nearest taxon index (betaNTI; B) in grassland and shrub-encroached soils.** GS: control grassland soil; SS: shrub-encroached soil.

## DISCUSSION

In this study, encroachment triggered significant changes in soil bacterial community composition (Fig. 3), and an apparent increase in bacterial alpha-diversity (Fig. 1), which is consistent with other studies showing that aboveground vegetation triggers a profound influence on belowground bacterial communities (*Bragazza et al., 2015*; *Gellie et al., 2017*). Recently, we found dramatic shifts in soil bacterial communities associated with shrub encroachment relative to distant grassland soils (i.e., >500 m;

**Table 2 Mantel test showing the effect of soil properties on bacterial community composition.**

| Variables | Mantel test | |
|---|---|---|
| | r | P |
| Soil pH | **0.295** | **0.003** |
| Soil moisture (%) | **0.218** | **0.012** |
| $NH_4^+$-N (mg/kg) | 0.198 | 0.053 |
| $NO_3^-$-N (mg/kg) | **0.282** | **0.017** |
| Dissolved organic C (mg/kg) | **0.295** | **0.006** |
| Dissolved organic N (mg/kg) | 0.090 | 0.183 |
| Total carbon (mg/g) | **0.196** | **0.049** |
| Total nitrogen (mg/g) | 0.114 | 0.164 |
| Total phosphorus (mg/g) | **0.185** | **0.041** |
| Soil inorganic carbon (mg/g) | 0.168 | 0.079 |
| Soil organic carbon (mg/g) | **0.412** | **0.002** |
| Spatial distance (m) | 0.068 | 0.181 |

Note:
Comparing differences between samples in bacterial community composition to differences between samples in variables (i.e., soil properties and spatial distance) by Mantel tests. Significant correlations are shown in bold ($P < 0.05$).

Xiang et al., 2018), which is consistent with the current study, which shows a restructuring of bacterial communities between shrub-encroached and adjacent (one m apart) grassland soils, indicating that soil bacterial community appears to be sensitive indicator of plant cover type. In addition, bacterial alpha-diversity showed significant correlations with soil nutrient levels (e.g., SOC, etc; Table 1), which increased following shrub encroachment (Table S1; Bragazza et al., 2015), indicating that elevated soil nutrients might reduce competition within bacterial communities and allow rare species to persist, leading to an increase in soil bacterial alpha-diversity (Xiang et al., 2018). Our results go beyond these findings by showing that the predicted metabolic function differed significantly between grassland and shrub-encroached soils (Fig. 5; Fig. S6), suggesting that shrub encroachment likely triggers significant shifts in grassland ecosystem functioning.

Similarly, we found strong evidence for reproducible environmental filtering in encroached and control soils in this study (Fig. 4), indicating that different vegetation types were associated with specific belowground bacterial communities (Wallenstein, McMahon & Schimel, 2007; Chu et al., 2016). Environmental filtering may include access to specific carbon sources and changes in soil chemistry (Prescott & Grayston, 2013). Previous research also showed substantial differences in bacterial community compositions among four vegetation types (Gibbons et al., 2017), providing evidence for dynamic and complex feedbacks between aboveground plant and belowground bacterial community structure (Shi et al., 2015; Gibbons et al., 2017).

Soil pH has been demonstrated to be a dominant factor in driving belowground bacterial community composition (Baker et al., 2009). However, compared to adjacent grassland soils (one m apart), shrub encroachment was not predominantly related to the shift in soil pH. The primary influence of pH on bacterial community composition was not

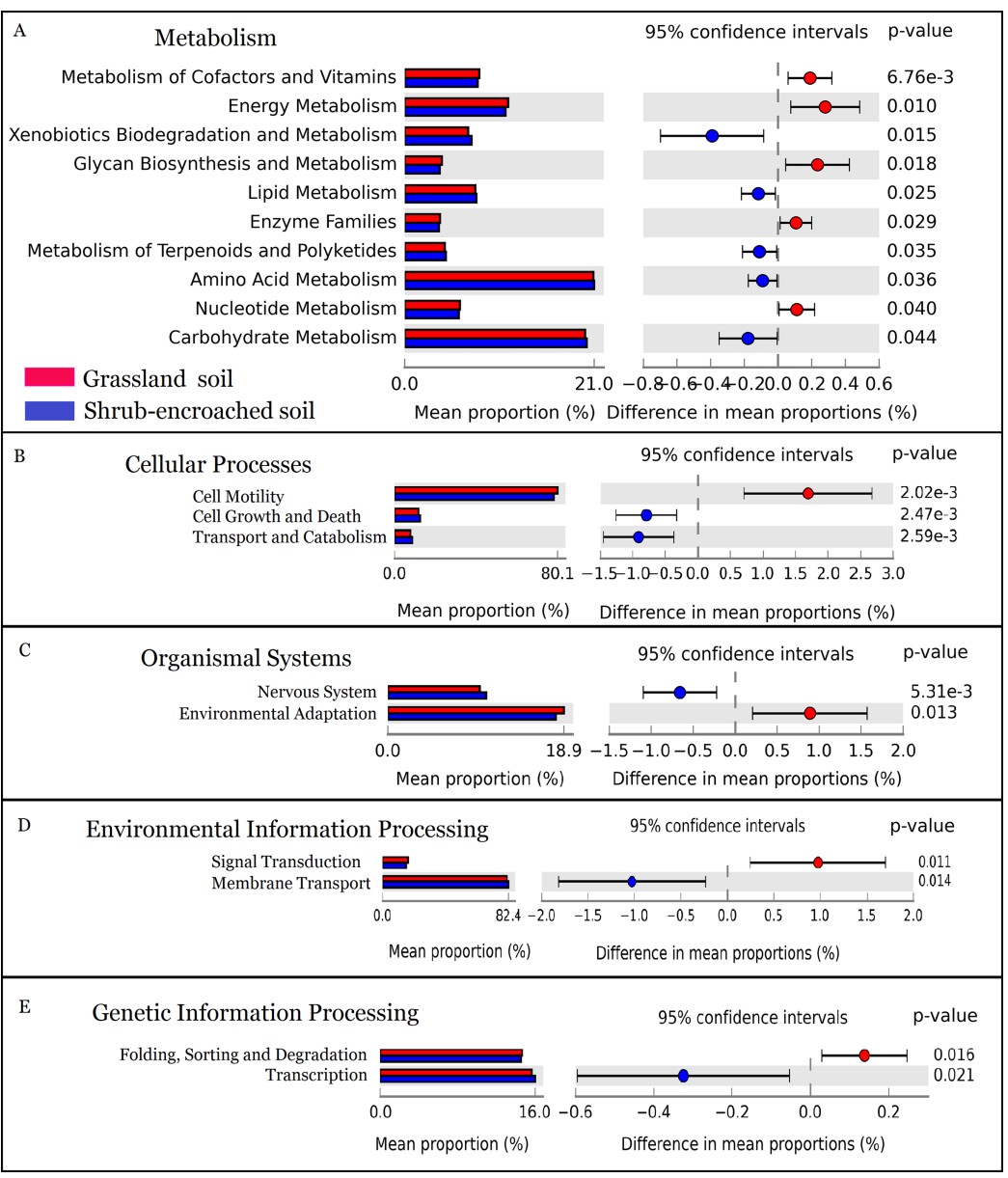

**Figure 5 The predicted metabolic function profiles of bacterial community.** Variation of metabolic function profiles of bacterial community in grassland and shrub-encroached soils analyzed by PICRUSt. (A) Metabolism; (B) Cellular Processes; (C) Organismal Systems; (D) Environmental Information Processing; (E) Genetic Information Processing.

detectable in this study, possibly induced by limited variation of pH range between grassland and nearby shrub-encroached soils. In this study, shrub-encroached soil was strongly related to an increase in SOC content, which was the primary factor in explaining the variance in bacterial community composition across sites (Table 2; Fig. S5). *Sul et al. (2013)* also demonstrated that SOC was the most important factor to explain the differences in the bacterial community composition in a tropical agricultural ecosystem. A prior study showed that plant communities altered SOC concentrations to indirectly affect belowground bacterial community composition (*Liu et al., 2014*). In addition,

soil carbon fraction might be a crucial factor in shaping microbial communities (*Zhou et al., 2012*). Plants may influence bacterial communities by determining the quantity and quality of the litterfall supply (*Wallenstein, McMahon & Schimel, 2007*) and/or by releasing photosynthetic products into the soil (*Staddon et al., 2003*). Shrubs may contribute qualitatively different carbon inputs (i.e., litterfall, root exudate, etc; *Schlesinger et al., 1990*; *Archer, Schimel & Holland, 1995*) to soils and thereby influence soil bacterial communities. Therefore, we speculate that shrub-mediated changes in SOC appear to be primary responsible for changes in composition of bacterial community.

A prior study demonstrated that shrub expansion was associated with enhanced N availability, which in turn facilitated shrub expansion and increased shrub patch density (*Chu & Gorgan, 2010*). We found that soil $NO_3^-$ content showed significant enrichment in shrub-encroached sites (Table S1). Moreover, shrub encroachment was related to elevated relative abundance of *Nitrospira*, which performs soil nitrification process (*Daims et al., 2015*) (Fig. 2), indicating that the higher relative abundance of *Nitrospira* might lead to the accumulation of soil $NO_3^-$ following shrub encroachment (*Xiang et al., 2018*). Soil $NH_4^+$ concentrations did not differ between grassland and shrub-encroached sites. Thus, enhanced N availability in shrub encroached sites appears to be induced by elevated soil $NO_3^-$, which may act as a positive feedback on shrub encroachment (*Chu & Gorgan, 2010*).

Overall, we propose a possible feedback among vegetation, soil properties, and bacterial community following encroachment based on our results, whereby: (1) shrub encroachment increases soil organic matter (e.g., litterfall, etc; *Schlesinger & Pilmanis, 1998*; *Kurc & Small, 2004*), which (2) activates soil microbes and alters soil nutrient cycling, and (3) greater resulting N availability facilitates shrub expansion and increased shrub densities around established shrub patches (*Chu & Gorgan, 2010*).

## CONCLUSIONS

This study demonstrated that shrub-encroached soils were associated with significant increase in bacterial alpha-diversity and dramatic restructuring of bacterial community composition. Environmental filtering (e.g., SOC content, etc) appears to mediate the influence of vegetation type on belowground microbial communities. The results of predicted metabolic function suggested that shrub encroachment might trigger large-scale disruptions of grassland ecosystem functioning. This work helps to further refine our knowledge of how shrub encroachment affects bacterial community structure in grassland ecosystems. However, we did not investigate the effect of encroachment on soil fungal communities, which might be more important for carbon cycling and closely related to changes in vegetation. This limitation should be addressed in future studies.

## ACKNOWLEDGEMENTS

We thank Ms. Kunkun Fan and Ms. Maomao Feng from Institute of Soil Science, Chinese Academy of Sciences, for assistance in data analysis.

### Funding

This work was supported by the National Natural Science Foundation of China (31801989, 31330012), Natural Science Foundation of Education Committee of Anhui Province (KJ2018A0001), the Training Project for Xingjia Xiang (S020118002, Z010139012) and the Strategic Priority Research Program (Grant #XDB15010101) of Chinese Academy of Sciences. Sean M. Gibbons was supported by a Washington Research Foundation Distinguished Investigator Award and by startup funds from the Institute for Systems Biology. The funders had no role in study design, data collection and analysis, decision to publish, or preparation of the manuscript.

### Grant Disclosures

The following grant information was disclosed by the authors:
National Natural Science Foundation of China: 31801989, 31330012.
Natural Science Foundation of Education Committee of Anhui Province: KJ2018A0001.
Training Project for Xingjia Xiang: S020118002, Z010139012.
Strategic Priority Research Program of Chinese Academy of Sciences: #XDB15010101.
Washington Research Foundation Distinguished Investigator Award.
Startup funds from the Institute for Systems Biology.

### Competing Interests

The authors declare that they have no competing interests.

### Author Contributions

- Xingjia Xiang performed the experiments, analyzed the data, contributed reagents/materials/analysis tools, prepared figures and/or tables, authored or reviewed drafts of the paper, approved the final draft.
- Sean M. Gibbons analyzed the data, contributed reagents/materials/analysis tools, prepared figures and/or tables, authored or reviewed drafts of the paper, approved the final draft.
- He Li performed the experiments, analyzed the data, contributed reagents/materials/analysis tools, approved the final draft.
- Haihua Shen conceived and designed the experiments, approved the final draft.
- Haiyan Chu conceived and designed the experiments, contributed reagents/materials/analysis tools, prepared figures and/or tables, authored or reviewed drafts of the paper, approved the final draft.

### Data Availability

The raw data is available at the Sequence Read Archive (SRA) of NCBI under the accession number SRP136091.

## Supplemental Information

Supplemental information for this article can be found online at http://dx.doi.org/10.7717/peerj.7304#supplemental-information.

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
