# Peer review of "Proximate grassland and shrub-encroached sites show dramatic restructuring of soil bacterial communities"

_PeerJ, doi:10.7717/peerj.7304_

## Round 0.1 · original submission · Minor Revisions

Small-scale variations in microbial communities are a common phenomenon in soil environments, while the underlying ecological processes are less understood. The results obtained from the proximate grassland and shrub-encroached sites provide a great example in exploring above key questions. The comments and concerns, raised by two experts in this field, are needed to be addressed point-to-point.

·

Basic reporting

The authors investigated the effects of shrub-encroachment on the soil bacterial community structure by Illumina Miseq. Overall, the manuscript was written with professional standard English and easy to follow. The structure, figures and tables in the MS are sound and readable. However, the introduction seems lacking of few lines on the importance of microbes. One question is that why the authors did not investigate the shift in fungal community, which are more important in carbon cycles and close related to vegetation changes.
Personally, the text in result part could be more straightforward than by using sophisticated, but rather confusing sentences. e.g.

Experimental design

The aim of the MS was clear and the experimental design were sound and logical. Materials and methods section were well described in most of the parts, except in Statistical analysis, in which, lines of 136-143 were not well explained and clarified. What are the purposes to use NTI, betwNTI as well as Co-occurence network analysis. The interpretaton from these analysis in the result were somehow vague.

Validity of the findings

The conclusion seems not focusing the aim of the study. The listed 1),2) and 3) should be in the background information section rather than in the conclusion. Secondly, the suggestion (line 245) can be confirmed easily by further analysis for community function based on the OTU taxonomy via computationally Predicted Functional Metagenomes (PFMs).

Line 198-199. Is it correct interpretation? Encroachment caused bacterial community shift, not vice versa.

·

Basic reporting

The manuscript “Proximate grassland and shrub-encroached sites show dramatic restructuring of soil bacterial communities” compared the bacterial community difference at the levels of composition and phylogenetics, aiming to understand the effects of shrub encroachment on soil bacterial community in grassland ecosystem. The results showed that shrub soil bacterial diversity was significantly higher than grassland, and their community structure substantially differred. SOC played a key role in driving the soil bacterial community difference in shrub and grassland, but pH did not. The study provided interesting findings, which contributes to understand the belowground response to shrub expansion.

Experimental design

No comments.

Validity of the findings

Here are my concerns:
1. The introduction failed to appreciate previous studies about the association of shrub and grassland with soil microbial community.
2. Phylogenetic diversity was an important index in the study, but was not mentioned the calculating method in the method section.
3. Line 163-164 should go to the section of bacterial alpha-diversity.
4. Line 169 "more relative abundance" should be a higher relative abundance.
5. Line 182, I had difficulty in understanding the sentence, there was a language problem.
6. SOC was demonstrated to be a key factor to differ the bacterial community in shrub and grassland soils, but authors did not carefully discuss this point. I'd encourage the authors discuss how SOC type and contents from shrub and grass affect bacterial communities.

Additional comments

See above.

---

## Round 0.2 · accepted · Accept

This is an important study showing the contrasting ecological processes underlying the microbial communities in proximate grassland and shrub-encroached sites.